# Empirical Studies on the Convergence of Feature Spaces in Deep Learning

## Abstract

While deep learning is effective to learn features/representations from data, the distributions of samples in feature spaces learned by various architectures for different training tasks, e.g., latent layers of Autoencoders (AEs) and feature vectors in Convolutional Neural Network (CNN) classifiers, have not been well-studied or compared. We hypothesize that the feature spaces of networks trained by various architectures (AEs or CNNs) and tasks (supervised, unsupervised, or self-supervised learning) share some common subspaces, no matter what types of architectures or whether the labels have been used in feature learning. To test our hypothesis, through Singular Value Decomposition (SVD) of feature vectors, we demonstrate that one could linearly project the feature vectors of the same group of samples to a similar distribution, where the distribution is represented as the top left singular vector (i.e., principal subspace of feature vectors), namely $\mathcal{P}$-vector. We further assess the convergence of feature space learning using angles between $\mathcal{P}$-vectors obtained from the well-trained model and its checkpoint per epoch during the learning procedure, where a quasi-monotonic converging trend from nearly orthogonal to smaller angles (e.g., $10°$) has been observed. Finally, we carry out case studies to connect $\mathcal{P}$-vectors to the data distribution, and generalization performance. Extensive experiments with practically-used Multi-Layer Perceptron (MLP), AE and CNN architectures for classification, image reconstruction, and self-supervised learning tasks on MNIST, CIFAR-10 and CIFAR-100 datasets have been done to support our claims with solid evidences.

## 1 Introduction

Blessed by the capacities of feature learning, deep neural networks (DNNs) (LeCun et al., 2015) have been widely used to perform learning tasks, ranging from classification, to generation (Goodfellow et al., 2014; Radford et al., 2015), in various settings (e.g., supervised, unsupervised, and self-supervised learning). To better analyze the features learned by deep models, numerous works have studied on interpreting the features spaces of the well-trained models (Simonyan et al., 2013; White, 2016; Zhu et al., 2016; Bau et al., 2017; 2019; Jahanian et al., 2020; Zhang & Wu, 2020).

**Invariance beyond the use of architectures and labels.** While existing studies primarily focus on the interpolation of a given model to discover mappings from the feature space to outputs of the model (e.g., classification (Bau et al., 2017) and generation (Jahanian et al., 2020)), the work is so few that compares the feature spaces learned by deep models of varying architectures (e.g., MLP/CNN classifiers versus Autoencoders) for different learning paradigms (Chen et al., 2020; Khosla et al., 2020; Spinner et al., 2018). More specifically, we are particularly interested in whether there exists certain *"statistical invariance"* in the feature space, no matter what type of architectures or whether label information (e.g., supervised vs. unsupervised vs. self-supervised (Chen et al., 2020) learning) are used in feature learning with the same training dataset.

**Hypotheses.** It is not difficult to imagine that the feature spaces of well-trained DNN classifiers in supervised learning setting might share some linear subspace (Vaswani et al., 2018). When models are well fitted to the same training set, the feature vectors of training samples should be projected to the ground-truth labels after a Fully-Connected Layer (i.e., a linear transform), while such linear subspace are supposed to distribute samples in a discriminative manner. We doubt that such

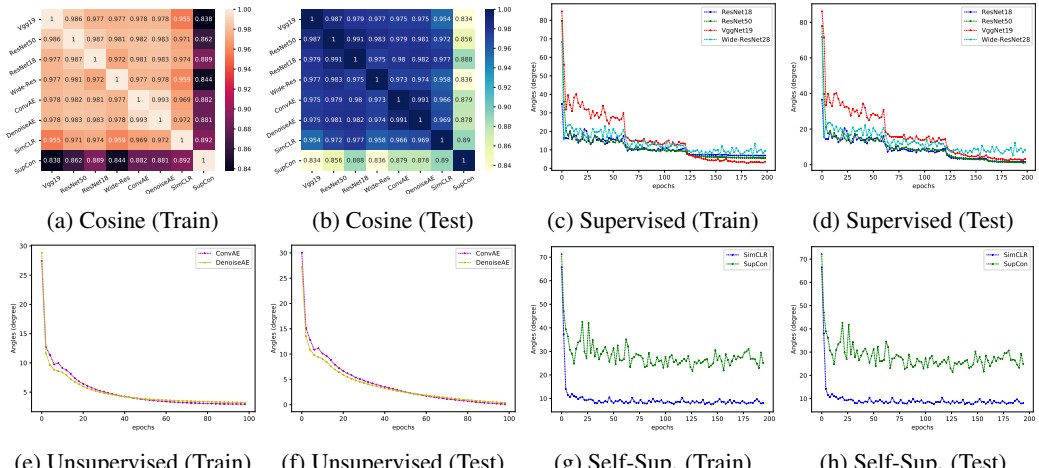

(a) Cosine (Train)  (b) Cosine (Test)  (c) Supervised (Train)  (d) Supervised (Test)

(e) Unsupervised (Train)  (f) Unsupervised (Test)  (g) Self-Sup. (Train)  (h) Self-Sup. (Test)

Figure 1: **The Common Feature Subspace and Converging Trends of $\mathcal{P}$-vector Angles with CIFAR-10.** Figure 1. (a)–(b) present cosine (in the range of [0,1]) of angles between the $\mathcal{P}$-vectors of well-trained models of various architectures under different learning paradigms, using training and testing datasets respectively. *A well-trained model here is the one trained under the suggested settings after 200 epochs for supervised/self-supervised CNN classifiers and 100 epochs for unsupervised AEs.* Figure 1. (c)–(h) present angles of $\mathcal{P}$-vectors between the well-trained model and its checkpoint per training epoch of three learning paradigms, where the converging trends of $\mathcal{P}$-vector angles from nearly-orthogonal to smaller ones have been observed in all models, no matter whether the feature extractors of these models are trained with / without labels. Note that we carried out experiments with different random seeds in 5 independent trials to obtain the averaged results above. More discussion are provided in Section 4.

subspace might be not only shared by supervised learners but also with AEs which are trained to reconstruct input data without any label information in an unsupervised manner, or even shared with self-supervised DNN classifiers (e.g., SimCLR (Chen et al., 2020)) which train (1) CNN feature extractor (using contrastive loss without labels) and (2) linear classifiers (using discriminative loss based on labels) separately in an ad-hoc manner. More specifically, we hypothesize that (H.I:) there exists certain common feature subspaces shared by well-trained deep models using the same training dataset, even though the architectures (MLPs, CNNs, and AEs) and the learning paradigms (supervised, unsupervised, and self-supervised) are significantly different. Further, as the training procedure usually initializes the DNN models from random weights and learns features from the training set step-by-step, we hypothesize that (H.II:) the training procedure gradually shapes the feature subspace over training iterations and asymptotically converge to the common subspace in certain statistical measure. Finally, we hypothesize that (H.III) the convergence to the common feature subspace would connect to the data distribution and performance of models, as such behavior indicates how well the features are learned from data. This hypothesis is motivated by the observation that when the DNN model tends to be linear the DNN feature subspace should be close to the data subspace, while the well-trained DNN models should be locally linear (Zhang & Wu, 2020) or piecewise linear (Arora et al., 2018).

**Contributions.** To test above three hypotheses, this work makes contributions in proposing new measures to the DNN features, namely $\mathcal{P}$-vectors, and conducting extensive experiments for empirical studies. We train deep models using various DNN architectures, multiple learning paradigms, and datasets, with the checkpoint restored per epoch. Then, we extract the feature vectors for either training or testing sample sets, from the model (Please see Section 3 for details) and discover some interesting relationships or associations as discussed below.

**I. $\mathcal{P}$-vector and Convergence:** Given the matrix of feature vectors (#samples[1] ×#features) for either training or testing samples, we perform the singular value decomposition (SVD) to obtain *left* and *right* singular vectors, characterizing the subspaces that samples distribute and the projection of features to subspaces respectively. We observe that deep models well-trained using the same

---

[1]We follow the convenience that denotes # as the term "the number of" for short.

dataset share similar *top left-singular vectors* (referring to the principal subspace of feature vectors), namely $\mathcal{P}$-vector in this study. For example, Figure 1 (a)–(b). show the high cosine similarity (close to 1) between the $\mathcal{P}$-vectors of any two well-trained DNNs for eight different deep learning architectures/tasks. This observation well backups H.I, where the common subspace in feature spaces shared by models trained with different architectures/paradigms has been observed. With checkpoints obtained along the training procedure, we estimate the angle that compares $\mathcal{P}$-vectors of the checkpoint per epoch and the well-trained one. We find such angle decrease over the number of epochs (in an overall manner) and would converge[2] to smaller ones. For example, Figure 1 (c)–(h). demonstrate the consistent converging trends of angles between $\mathcal{P}$-vectors of the well-trained model and its checkpoints in progress of training for eight different deep learning architectures/tasks. This observation supports H.II, where convergence to the common feature subspace are expected.

**II. Data Distribution and Performance:** To connect to the data distributions, we intend to compare feature vectors and original data vectors of samples using $\mathcal{P}$-vectors. In addition to the $\mathcal{P}$-vector estimated from the feature vectors of a deep model (namely a "model $\mathcal{P}$-vector"), We simply form the sample vectors (either for training or testing) into a #samples×#data_dimension matrix and perform SVD to obtain the top left singular vector as the $\mathcal{P}$-vector of samples (namely a "data $\mathcal{P}$-vector"). We estimate the angles between the model and data $\mathcal{P}$-vectors, and find a trend of convergence (from nearly orthogonal to relatively small angles), where we can see the well-trained models would incorporate smaller angles than ones in the early stage of training processes. We further correlate such angles with training and testing accuracy of the models, where we observe significant negative correlations in most cases of experiments. The trends show that the model with a smaller angle between the model and data $\mathcal{P}$-vectors would enjoy better performance. The evidences backup H.III.

## 2 RELATED WORK

In this section, we first present the preliminaries in understanding feature learning of DNNs, then discuss the most relevant works to our studies.

As early as 2013, (Simonyan et al., 2013) proposed to visualize the features learned by deep convolutional neural networks (CNN) and made sense of discriminative learning via deep feature extraction. For generative models, (White, 2016; Zhu et al., 2016) studied the interpolation of latent spaces while (Zhu et al., 2016) discovered an user-controlled way to manipulate the images generated through the surrogation of latent spaces via manifolds. Later, (Bau et al., 2017) presented the visual concepts learned in the feature spaces of discriminative models through network dissection on specific datasets while the same group of researchers also proposed GAN dissection (Bau et al., 2019) – an interactive way to manipulate the semantics and style of image synthesis. (Richardson & Weiss, 2018) compared GAN and Gaussian Mixture Models (GMMs) to understand the capacity of distribution learning in GAN. (Berthelot et al., 2019) proposed to improve understanding and interpolation of Autoencoders using adversarial regularizer while (Spinner et al., 2018) compared AEs with its variational derivatives to interpret the latent spaces. More recently, (Saxe et al., 2019) mathematically analyzed the process of neural representation construction from the perspectives of learning dynamics of deep neural networks. (Jahanian et al., 2020) studied the "steerability" of GAN, where it discovered point-to-point editing paths for content/style manipulation. (Zhang & Wu, 2020) uncovered the phenomena that DNN classifiers with piecewise linear activation tend to map the input data to linear subregions. Other impressive studies in this line of research include (Nguyen et al., 2016; Arvanitidis et al., 2018; Sercu et al., 2019).

**Discussion.**  The most relevant studies to our work are (Bau et al., 2017; Zhang & Wu, 2020; Saxe et al., 2019; Lee et al., 2019). For discriminative models, (Bau et al., 2017) recovered visual features learned by CNN classifiers with a priorly labeled dataset, and quantified then compared the feature learning capacities (namely "interpretability" in the work) of different DNN models through patterns matching with the ground truth. Compared to (Bau et al., 2017), we carry out the empirical studies on a wide range of datasets without any prior information on their features and observe consistent phenomena in the distribution of samples in the feature spaces. Furthermore, while (Zhang & Wu, 2020)

---

[2]In our research, we name convergence as the decreasing trend of $\mathcal{P}$-vector angles from a larger one to a smaller one (e.g., $10°$ for supervised CNN classifiers and SimCLR, $\leq 10°$ for ConvAEs/DenoiseAEs, and $30°$ for SupCon) over training epochs. For reference, Cosine($10°$)=0.985 is close to 1.0

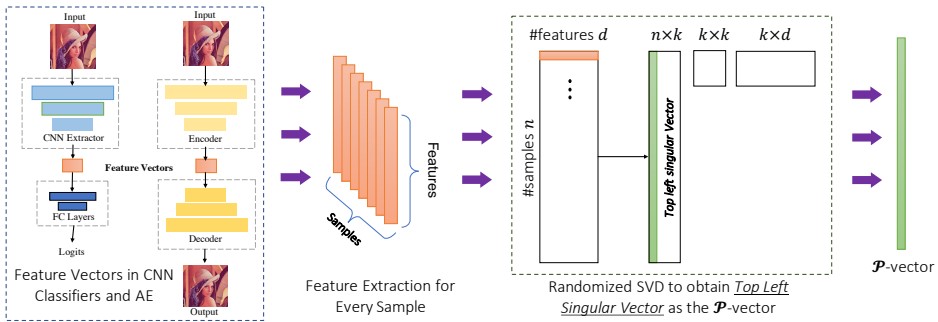

Figure 2: Obtain $\mathcal{P}$-vectors of deep neural networks using a group of samples

studied the properties of regions where a supervised DNN classifier with piecewise linear activation behaves linearly, our work observes the common linear subspaces shared by the features learned by the networks that are trained with different architectures (e.g., MLP/CNN classifiers and AEs with ReLU activation) and paradigms (e.g., supervised, unsupervised, and self-supervised learning). Furthermore, both our work and (Saxe et al., 2019) propose to compare feature/representation learning by various models through SVD, while we perform SVD to investigate the distribution of samples in feature space and (Saxe et al., 2019) uncovered the latent structures in input-and-output of neurons.

To understand the latent space of generative models, some statistical/mathematical tools (Richardson & Weiss, 2018; Lee et al., 2019) have been introduced. Through comparisons between GAN and GMM, (Richardson & Weiss, 2018) uncovered GAN's superiority in feature/texture reconstruction and in the meanwhile its incompetence in distribution learning, while (Lee et al., 2019) provides some analytical insights on the structure of latent spaces. In our work, we propose $\mathcal{P}$-vectors to model the distribution of samples in the feature/latent spaces of DNN classifiers and AEs. To the best of our knowledge, we make unique contributions compared to the above work.

## 3   METHODOLOGIES: FEATURE VECTORS, SINGULAR VALUE DECOMPOSITION, AND $\mathcal{P}$-VECTORS

As mentioned in Section. 1, we carry out extensive experiments on analyzing the feature vectors of DNN models with various architectures and different training paradigms through the newly-proposed $\mathcal{P}$-vectors. In this section, we present the design of experiments in details, where we cover three key procedures of the experiment as shown in Figure 2.

**Feature Vector Extraction.**    Given a model, either the well-trained one or a checkpoint obtained during the training process, we extract the feature vector for every sample, with respect to the architectures. For DNN classifiers (either under supervised/self-supervised learning), we use the output of CNN feature extractor (i.e., the input to the Fully-Connected Layer) as the feature vector of the given sample, while we vectorize the output bottleneck layer as the feature vector for AEs. Note that, in our research, we consider AEs with symmetric architectures of encoders and decoders only.

**Singular Value Decomposition with Feature Vectors.**    Given the feature vector for every sample, we form a #samples×#features matrix and perform SVD to obtain the top left singular vector as the $\mathcal{P}$-vector. Furthermore, there is no need to solve the singular vectors of the complete spectrum, as only the top singular vector is requested for $\mathcal{P}$-vector estimation. In this way, we propose to use Randomized SVD (Halko et al., 2011) that compresses the feature domain and approximate the low-rank structure of SVD for acceleration purpose. Actually, we compare the numerical solution of Randomized SVD and Common SVD for ResNet-50 on CIFAR-10 dataset (#features=256 and #samples= 50,000), where we need to perform SVD on a #samples×#features matrix and the $\mathcal{P}$-vector/top left singular vector should be with dimensions. Compared to the vanilla SVD, around 109x (from 44.88 seconds to 0.41 seconds) speedup has been achieved by Randomized SVD while no significant numerical errors having been found in the results.

**$\mathcal{P}$-vectors, Principal Subspace, and the Distribution of Samples in Feature Spaces.**    The top left singular vector of #samples×#feature matrix represents the *principal subspace* (Jolliffe, 2002) that samples are distributed into a one-dimensional space (with the maximal variances) by the projection of principal component (Abdi & Williams, 2010) of features. In this way, we can compare

and measure the divergence between the feature spaces of two DNNs through assessing the angle between their $\mathcal{P}$-vectors based on the same set of samples.

**Discussion.** We plan to discuss the properties or $\mathcal{P}$-vectors from the aspects as follow.

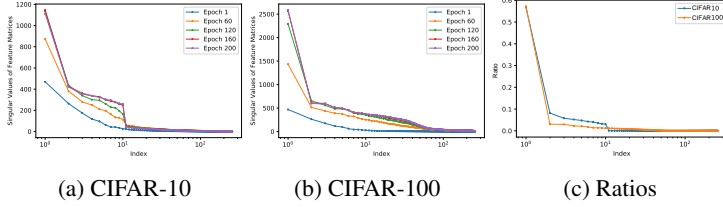

(a) CIFAR-10       (b) CIFAR-100       (c) Ratios

Figure 3: Singular Value Distribution and Explained Variance Ratios of the Matrix of Feature Vectors

*Significance of $\mathcal{P}$-vectors.* A possible threat to validity of using $\mathcal{P}$-vectors to analyze the feature space of DNN is that $\mathcal{P}$-vectors might fail to capture the necessary information to represent the features learned. To validate the capacity of $\mathcal{P}$-vectors to represent the feature space, we carried out Singular Value Decomposition on the matrix of feature vectors, using CIFAR-10 and CIFAR-100 datasets both based on ResNet-50 models, and obtain the distribution of singular values over indices. More specifically, we compute the distributions of singular values for the feature matrices obtained in the $1^{st}$, $60^{th}$, $120^{th}$, $160^{th}$ and $200^{th}$ epochs to monitor the change of singular value distributions throughout the training procedure. It has been observed in Figures 3(a) and (b) that a "cliff" pattern in the distribution becomes more and more significant after epochs of training for both CIFAR-10 and CIFAR-100 datasets – a very small number (less than 10) of top singular values might dominate the whole distribution. In Figure 3(c), we plot the curve of explained variance ratio $\sigma_k^2 / \sum_{j=1}^{d} \sigma_j^2$ for every pair of singular vectors, using well-trained models of 200 epochs based on CIFAR-10 and CIFAR-100, where $\sigma_k$ refers to the $k^{th}$ singular value and $d$ is the rank of matrix. The explained variance ratio of the top-1 singular vectors (i.e., the $\mathcal{P}$-vector and the top-1 right singular vector) is more than 50% while the second top singular vectors are less than 10%. Results show the use of $\mathcal{P}$-vectors could represent the features learned. In addition to the use of top-1 singular vectors (the $\mathcal{P}$-vector), in Appendix (A.7), we also discuss the results of including more singular vectors in analysis, where no consistent observations have been obtained.

*Distribution of Values in a $\mathcal{P}$-vector.* As a singular vector, a $\mathcal{P}$-vector should be a unit-vector, where the sum of squared value on every dimension of the vector should be 1. To take a closer look at the $\mathcal{P}$-vector, we first retrieve the value on every dimension of the vector and count the frequency that each value appears. We normalize the frequency with the

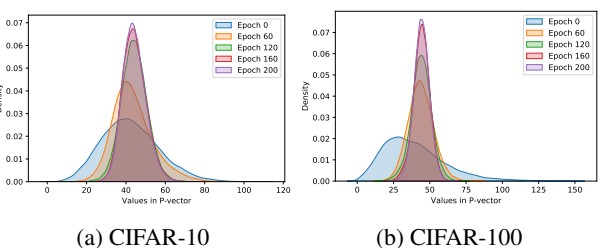

(a) CIFAR-10            (b) CIFAR-100

Figure 4: Distribution of Values in the $\mathcal{P}$-Vector

total counts and plot the smoothed probability density of the distributions (of the values in the $\mathcal{P}$-vector) in Figure 4 (a) and (b) for CIFAR-10 and CIFAR-100 datasets. Specifically, we plot the distribution for $\mathcal{P}$-vectors obtained in different epochs throughout the training procedure, where a clear "concentration" process could be observed. In the beginning, probably due to the random initialization, the values are flatten in a wider range. With the training epochs, the distributions in the both figures would be "concentrated" into narrow ones with reduced ranges. We could observe the peak shifts over training epochs in both figures, while the distributions based on two datasets are significant different from both magnitudes and ranges' perspectives. Examples on the raw frequency of values in $\mathcal{P}$-vector is included in Appendix (A.6), where same trends could be observed. Note that two $\mathcal{P}$-vectors are not necessary to be close, when their distributions (of values) are close. Because the specific value assigned to every sample in the $\mathcal{P}$-vector could be significantly different. Thus, analyzing the distribution of values in $\mathcal{P}$-vectors with respect to the distribution of data might be a part of future work.

*Ways to compare $\mathcal{P}$-Vector.* Given the same set of samples, $\mathcal{P}$-vectors of the two DNNs should be in the equal length, as they are both the top singular vectors in the sample side. Furthermore, we can easily measure their divergence via Cosine of two vectors. A larger cosine of two vectors (e.g., close to 1.0) usually refers to the evidence that the two networks share subspace in their features. Note that, in high-dimensional spaces, the chance of orthogonality between two random vectors

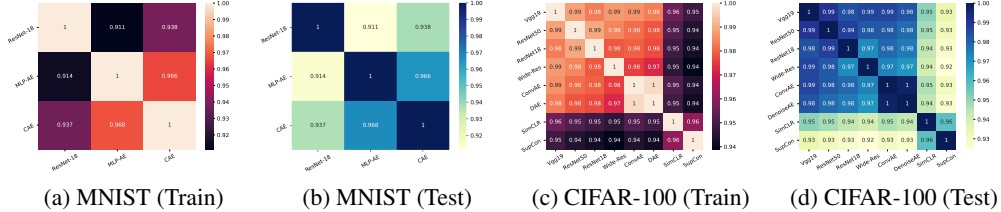

| (a) MNIST (Train) | (b) MNIST (Test) | (c) CIFAR-100 (Train) | (d) CIFAR-100 (Test) |

Figure 5: Cosine of angles between the $\mathcal{P}$-vectors of well-trained models

appears more frequently, due to the curse of dimensionality (Pestov, 1999). Thus, given a sample set such as CIFAR-10 with more than 60,000 samples, when the cosine measure close to 1.0 or the angle between the two $\mathcal{P}$-vectors (with 60,000 dimensions) is small, we can conclude that the two networks would share a subspace in the feature spaces in high confidence. Of-course, there might exist other ways to perform analysis using $\mathcal{P}$-vectors. In our future work, we plan to leverage advanced numerical tools Björck & Golub (1973) to estimate the angles between the subspaces of DNN models in general dimensions.

# 4 UNCOVERING COMMON FEATURE SUBSPACE WITH $\mathcal{P}$-VECTORS

**Common Subspaces.** To demonstrate the existence of common subspaces, we propose to measure the angles between the $\mathcal{P}$-vectors extracted from well-trained models using the same datasets. For any two models, a small angle between the two $\mathcal{P}$-vectors refers to the facts that the two models would project these samples to similar principal subspaces in feature spaces, as the $\mathcal{P}$-vectors are estimated as the top left singular vector of the feature matrix (i.e., #samples×#features) SVD based on the same group of samples. Figure. 1 (a)–(b) presents the cosine of angles between the $\mathcal{P}$-vectors of well-trained models (trained with various architectures and tasks using CIFAR-10 dataset). To generalize our observations, we propose to compare the $\mathcal{P}$-vectors of models using the feature matrices based on the training and testing samples in Figures. 1 (a)

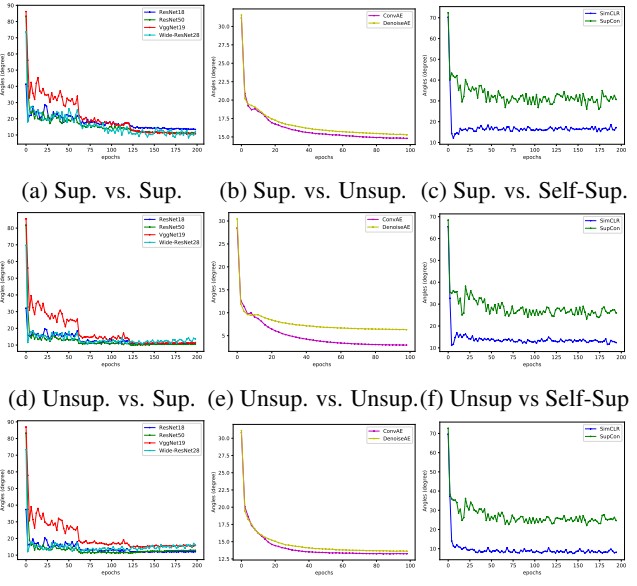

(a) Sup. vs. Sup. (b) Sup. vs. Unsup. (c) Sup. vs. Self-Sup.

(d) Unsup. vs. Sup. (e) Unsup. vs. Unsup.(f) Unsup vs Self-Sup

(g) Self. vs Sup. (h) Self. vs Unsup. (i) Self. vs Self-Sup.

Figure 6: $\mathcal{P}$-vector angles between checkpoint per epoch and well-trained models using CIFAR-10.

and (b) respectively. We carry out the same experiments on CIFAR-10 and CIFAR-100 datasets, and present the cosine of angles between the $\mathcal{P}$-vectors of well-trained models in Figure 5. All experiments demonstrate that, with a relatively large cosine (close to 1), the models well-trained using the same dataset, no matter what types of architectures or whether labels has been used in the feature learning of various tasks, share a common subspace.

**Converge to the Common Subspace** To understand the dynamics of feature learning that shapes the common subspace from scratch, we measure the change of angles over epochs between the $\mathcal{P}$-vectors of training models (i.e., checkpoint in the epoch) and well-trained ones in comparisons. In Figure 1 (c)–(h), for every model of various tasks, we present angles between the $\mathcal{P}$-vectors of the well-trained model and itself's training checkpoint per epoch. A consistent convergence could be observed. As the $\mathcal{P}$-vectors of well-trained models are close to each others (see in Figures 1 and 5), we can conclude the feature vectors extracted by these models would evolve on time and gradually converge to share the common subspace during the learning procedure.

We carry out a more comprehensive model-to-model comparison using CIFAR-10 dataset. Figures. 6 (a)–(c) present the convergence of $\mathcal{P}$-vector angles between the well-trained supervised models and the checkpoints per epoch of supervised, unsupervised, and self-supervised learning models, where we use the well-trained Wide-ResNet28 (trained with 200 epochs under suggest settings) as the reference of supervised models. Figures. 6 (d)–(f) present $\mathcal{P}$-vector angles between the well-trained ConvAE (trained with 100 epochs under suggest settings as the reference of unsupervised learning) and the checkpoints per epoch of supervised, unsupervised, and self-supervised learning models. Figures. 6 (g)–(i) present the $\mathcal{P}$-vector angles between the well-trained SimCLR representations (trained under suggest settings (Chen et al., 2020) as the reference of self-supervised learning) and the checkpoints per epoch of supervised, unsupervised, and self-supervised learning models. The trends of convergence in all comparison further validate our hypotheses.

**Discussion.** To better visualize every comparison, we use the maximal angles achieved during the first epoch to represent the angles between $\mathcal{P}$-vectors corresponding to the first epoch in Figures 1 and 6. Actually, in the first epoch, there would incorporate some non-monotonic trends for the angles varying over the number of iterations. Figure 7 presents the angles between the $\mathcal{P}$-vectors of the

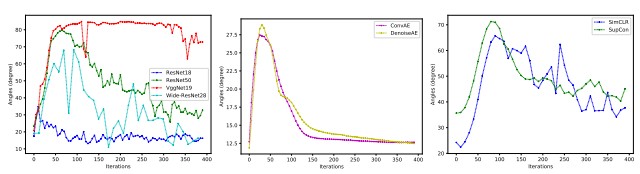

(a) Supervised     (b) Unsupervised     (c) Self-Sup.

Figure 7: Angles between the $\mathcal{P}$-vectors of the training and well-trained models over the number of iterations in the first epoch using CIFAR-10.

training and well-trained models over the number of iterations in the first epoch in three settings of learning, where we use CIFAR-10 dataset for the experiments. In this way, we recover the procedure that the learning procedure shapes the feature space – *with the random weight initialization, the angles of $\mathcal{P}$-vectors between the initial models and the well-trained models are large, the angles drop down and rebound in a zigzag trend quickly in all three learning paradigms. In the rest of learning procedure, the angles between the $\mathcal{P}$-vectors drop down and converge to smaller ones over the number of iterations.* (**H.II**)

Note that in Appendix (A.7), we include the comparison between the top singular vectors other than the $\mathcal{P}$-vectors, i.e., the angles of top-2, 3, 4, 5 and 6 left singular vectors between models, where we cannot observe the converging trends.

## 5   CASE STUDIES ON $\mathcal{P}$-VECTORS WITH DATA DISTRIBUTIONS AND MODEL PERFORMANCE

In this section, we intend to extend our observations to connect $\mathcal{P}$-vector with the (raw) data distributions and the generalization performance of models.

**Correlation to Data $\mathcal{P}$-vectors.** Given the raw data matrix, i.e., a #samples×#data dimensions matrix, we obtain the Data $\mathcal{P}$-vector[3] of these samples using the top left singular vector of the raw data matrix, which represents the principal subspace of data (or the position of every sample projected by the principal component of the data). To understand the connection between models and data, We

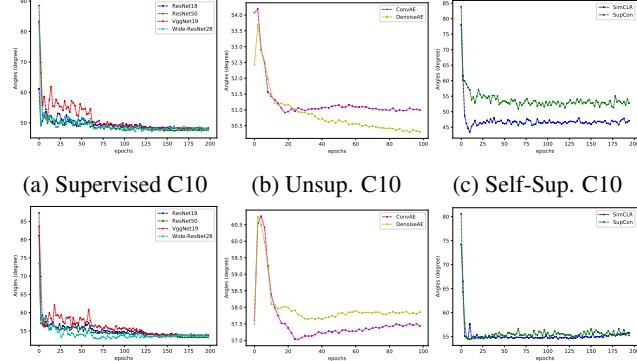

(a) Supervised C10    (b) Unsup. C10    (c) Self-Sup. C10

(d) Supervised C100    (e) Unsup. C100    (f) Self-Sup. C100

Figure 8: Angles between the model and data $\mathcal{P}$-vectors per training epoch. C10: CIFAR-10, C100: CIFAR-100

---

[3]We use the term "model $\mathcal{P}$-vector" to represent the $\mathcal{P}$-vector estimated using the feature vectors of a deep model, while using "data $\mathcal{P}$-vector" as the top left singular vector of the raw data matrix.

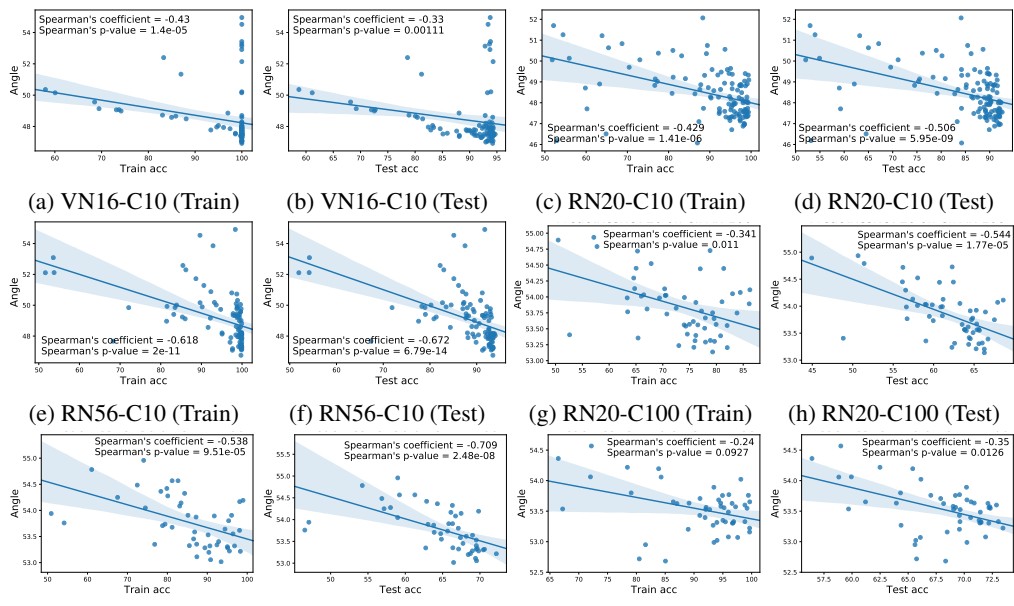

Figure 9: Strong and consistent correlations between the model performance (training and testing accuracy) and the angels between model and data $\mathcal{P}$-vectors using CIFAR-10 and CIFAR-100 datasets. VN: VggNet, RN: ResNet, C10: CIFAR-10, and C100: CIFAR-100.

carry out the case study using CIFAR-10 dataset, so as to measure the angles between the model $\mathcal{P}$-vector and the data $\mathcal{P}$-vector and how the angles change over the number of training epochs.

Figure.8 shows the convergence of $\mathcal{P}$-vector angles between the raw data and the checkpoints per epoch of supervised, unsupervised, and self-supervised learning models. The angles between the data and model $\mathcal{P}$-vectors start from about orthogonal and generally decrease and converge to about $50°$ degree to $60°$ degree. Note that the angle points corresponding to the first epoch on the curves are the largest $\mathcal{P}$-vector angles during the training procedure of the first epoch from the random scratch. The results indicate that the principal subspace of the well-trained models are more close to the principal subspace of the raw data, no matter what type of architectures is used for what kind of learning tasks and whether the labels are used for training.

**Correlation to Deep Learning Performance.** We further explore the relationship between the performance of the various models on different datasets and the angles between $\mathcal{P}$-vectors and data $\mathcal{P}$-vectors. Experiments are delivered using ResNet-20/56/110 and VGG-16 on CIFAR-10/100 datasets. As shown in Fig.9, there exists a strong and consistent correlation between the training/testing accuracy of models and the angles between the model and data $\mathcal{P}$-vectors. Note that in these experiments, we use the samples in the training dataset to estimate the model and data $\mathcal{P}$-vectors while *avoiding the use of validation information*, so as the demonstrate the connection of $\mathcal{P}$-vectors all based on training samples to the generalization performance. For CIFAR-100 dataset, we use ResNets with pre-activation enabled.

To avoid the dominance of some outliers, we conduct correlation analysis between the rank of model performance and the rank of angles by the use of Spearman's coefficients and p-values. With $0.05$ as the threshold for significance, we find significance in the correlations between the angles between model and data $\mathcal{P}$-vectors and the training/testing accuracy for all above cases, except the correlation between the angles and training accuracy on ResNet-100 using CIFAR-100 dataset (p-value=0.0927). In Appendix (A.8), we also include an additional correlation analysis based on log-log plots, where we can further validate our observation. In this way, we could conclude that (1) both training and testing accuracy are correlated to the angles between the model and data $\mathcal{P}$-vectors, (2) the strong correlations between angles and the testing accuracy might not be caused by the correlations between the angles and training accuracy, as the earlier ones are even stronger, (3) the angles between the model and data $\mathcal{P}$-vectors would be a reasonable performance indicator, as they are strongly, consistently, and significantly correlated to the testing accuracy. This observation coincides our intuition that a significant (locally) linear term exists in the well-trained model (Zhang & Wu, 2020), which makes DNN feature principal subspace correlate to the data principal subspace.

| Methods | Prediction Score |
|---|---|
| VC Dimension (Vapnik, 2013) | 0.020 |
| Jacobian norm w.r.t intermediate layers (Jiang et al.) | 2.061 |
| **Angles between the model and data $\mathcal{P}$-vectors** | **3.325** |
| Distance to Initialization (Nagarajan & Kolter, 2019) | 4.921 |
| **Distance from Initialization + Angles between the model and data $\mathcal{P}$-vectors** | **4.971** |
| Sharpness of the convergence point (Jiang et al., 2019) | 10.667 |
| Pseudo validation accuracy | 13.531 |
| **Pseudo validation accuracy + Angles between the model and data $\mathcal{P}$-vectors** | **15.618** |

Table 1: Scores of different method to predict the generalization gap using CIFAR-10 and SHVN.

**Applications to Generalization Performance Prediction.** To further demonstrate the feasibility of using the $\mathcal{P}$-vector as a "validation-free" measure of generalization performance, we use the experiment settings of "Predicting Generalization in Deep Learning Competition" at NeurIPS 2020 to evaluate $\mathcal{P}$-vectors on predicting the generalization performance of a wide range of models using the training dataset. The competition offers a large number of deep models trained with various hyper-parameters and DNN architectures, while the official evaluator for the competition first predicts the generalization performance of every model using the proposed measure, then verify the prediction results through the mutual information (the higher the better) between the proposed measures and the (observable) ground truth of generalization gaps.

In the experiments, we propose to use the angles between the model and data $\mathcal{P}$-vectors using the training dataset as the metrics of generalization performance. In the comparisons with the proposed $\mathcal{P}$-vectors, we include several baseline measures in generalization performance predictors, including VC Dimension (Vapnik, 2013), Jacobian norm w.r.t intermediate layers (Jiang et al.), Distance from the convergence point to initialization (Nagarajan & Kolter, 2019), and the Sharpness of convergence point (Jiang et al., 2019). In addition to these methods, we also propose "Pseudo Validation Accuracy" as a measure for comparisons, where this measure first uses random data augmentation apply to the original set of training data to generate a set of "pseudo validation samples", then tests the accuracy of the model using "pseudo validation samples".

Table 1 presents the comparisons between the proposed measures and baselines. It shows that when the proposed measure – angles between the model and data $\mathcal{P}$-vectors – stands alone, the measures significantly outperform the baseline methods including Jacobian norm w.r.t intermediate layers and the VC dimensions. However, through complementing with other metrics, the metrics based on $\mathcal{P}$-vector angles could be further improved in predicting the generalization performance and finally outperform all baseline methods when combining with "Pseudo validation accuracy". Note that we combine the results of two metrics through weighted aggregation (Pihur et al. (2009), with a constant weight 0.05) of two ranking lists that are sorted according to the two metrics respectively.

All above experiment results in three case studies backup our hypothesis that the $\mathcal{P}$-vectors of the model feature spaces would connect to the data distribution and performance of models (**H.III**).

# 6 CONCLUSION

In this work, we propose the $\mathcal{P}$-vector, defined as the top left singular vector of the feature matrix (the #samples×#features matrix) through SVD, to characterize the principal subspace of the feature space founded by the samples. We observe that, no matter what type of DNN architectures or whether the labels have been used to train the models, the angle of $\mathcal{P}$-vectors between any two models would decrease to smaller ones (e.g., around $10°$ for most models in this study and cosine($10°$)=0.985), when the models are well trained with the same dataset. We conclude that the feature spaces all well-trained DNN models using the same training dataset would share a principal subspace. Furthermore, during the training procedure from the random scratch, the model $\mathcal{P}$-vector would slowly approach to the data $\mathcal{P}$-vector (defined as the top left singular vector of the raw data matrix, i.e., #samples×#data dimensions), where the data and model $\mathcal{P}$-vectors start from an almost-orthogonal status while the angles between the data and model $\mathcal{P}$-vectors overall decrease and converge to smaller ones (e.g., $50°$–$60°$). Finally, we find that angles between the model and data $\mathcal{P}$-vectors are strongly correlated to the performance of models (i.e., both training/testing accuracy) while they are capable of predicting generalization performance, even when the model and data $\mathcal{P}$-vectors are all estimated using training dataset only. As was discussed, we believe the empirical observations obtained here are partially due to the local linearity of DNN models (Zhang & Wu, 2020); our future work may focus on the theoretical understanding to these phenomena.

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

# A APPENDIX

## A.1 COMPARISON OF ANGLES BETWEEN MODEL AND WELL-TRAINED $\mathcal{P}$-VECTOR USING DIFFERENT ARCHITECTURES ON CIFAR-100

In the main text, we presented the result on CIFAR-10 dataset. To generalize the observations, we repeated the experiments on CIFAR-100 dataset to validate our hypothesis of the convergence of the angles between model checkpoints and well-trained model $\mathcal{P}$-vectors. We investigate the change of angles over the $\mathcal{P}$-vectors of training model checkpoints per epoch with comparison to the $\mathcal{P}$-vectors of well-trained models (model of epoch 200 in our case). As shown in Fig.10, a gradually decreasing manner of the curves for the angle between $\mathcal{P}$-vectors and all angles between $\mathcal{P}$-vectors cross models with different supervisory manners generally converge to a value that smaller than $10°$ degree. We can conclude that the hypothesis of the existence of common subspace during the learning procedure also stands on the experiments with CIFAR-100 dataset.

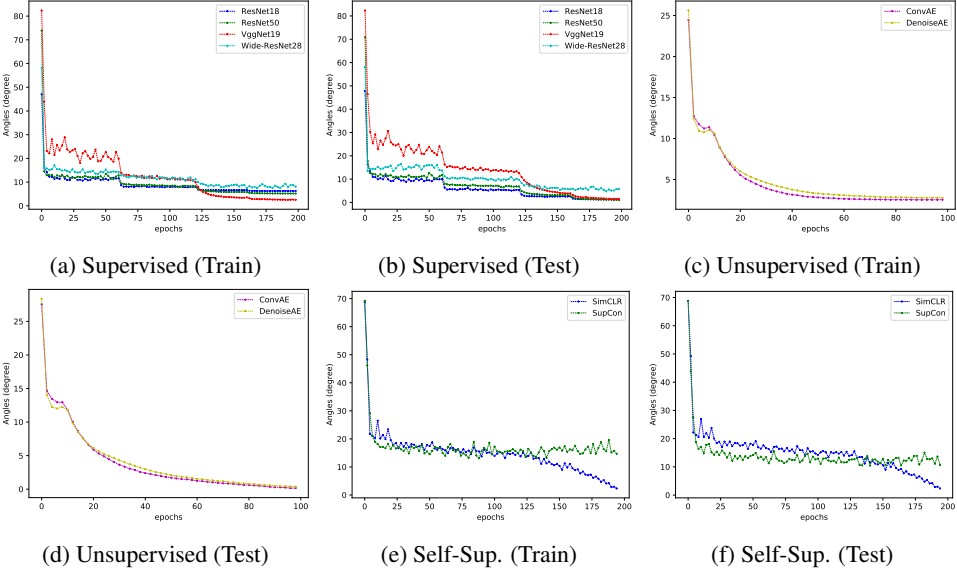

| (a) Supervised (Train) | (b) Supervised (Test) | (c) Unsupervised (Train) |
| (d) Unsupervised (Test) | (e) Self-Sup. (Train) | (f) Self-Sup. (Test) |

Figure 10: Convergence to the Common Feature Subspace with CIFAR-100. Curves of angles of $\mathcal{P}$-vectors between the well-trained model and its checkpoint per training epoch of three learning supervisory manners. The trends of convergence for the angles can be observed in all models.

## A.2 THE MON-MONOTONIC TREND WITHIN THE FIRST EPOCH

The variation of angles between $\mathcal{P}$-vectors for the well-trained model and its checkpoint per training epoch of each iteration in the first epoch. The non-monotonic trends within the first epoch also incorporate in the experiments on CIFAR-100 datasets. Fig.11 shows the curves indicating the variation of angles between the training model $\mathcal{P}$-vectors and the well-trained model $\mathcal{P}$-vectors in the iterations in the first epoch. As we use 128 as the batch size in training procedure, the number of iterations for updates is 391 per epoch. We obtain the observation of a non-monotonic trend that the angle first rises with the random initialization and drop down. And in the rest of training process, the angles keeps the approximately monotonically decreasing and converging to small values. The experiments shows consistent result and conclusion on CIFAR-100 dataset with the discussion in section 4.

## A.3 MODEL-TO-MODEL COMMON SUBSPACE

We also test and verify the model-to-model common subspace shared by models trained with different supervisory manners on CIFAR-100 dataset. Experiments carried out to evaluate the angles between $\mathcal{P}$-vectors for checkpoints of all models and $\mathcal{P}$-vectors for well-trained supervised, unsupervised and self-supervised models, where we use the well-trained Wide-ResNet28/Convolution

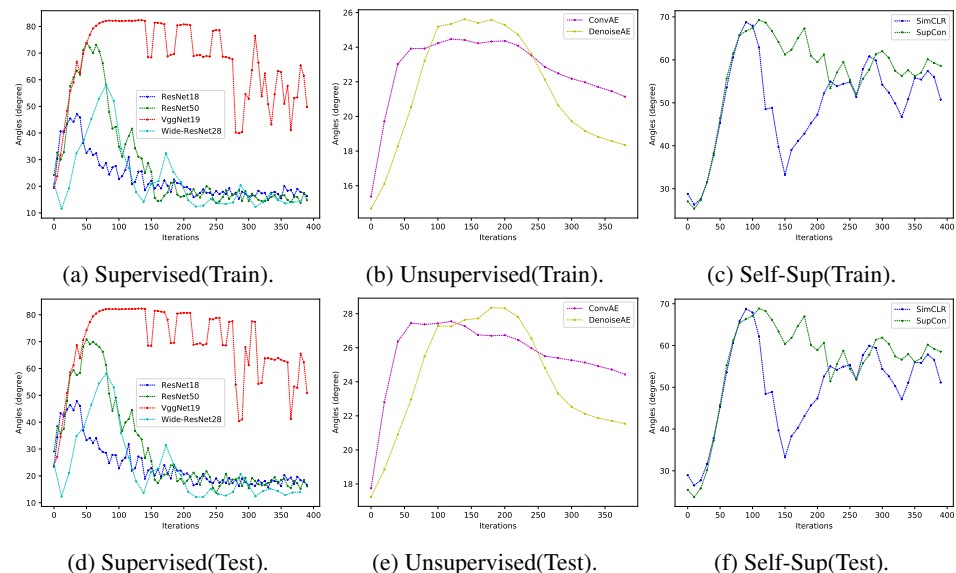

Figure 11: Angles between the $\mathcal{P}$-vectors of the training and well-trained models over the number of iterations in the first epoch using CIFAR-100.

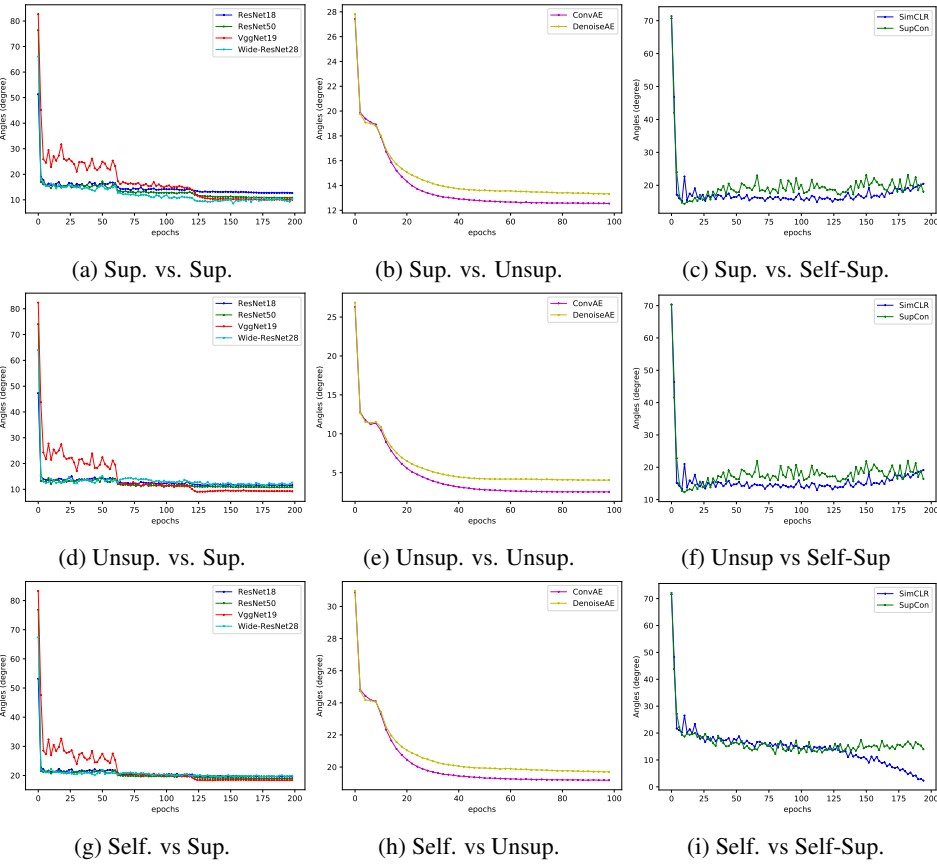

Figure 12: Convergence of $\mathcal{P}$-vector angles between checkpoint per epoch and well-trained models using CIFAR-100.

Auto-encoder and SimCLR model (trained with 200 epochs under suggest settings) as the reference of supervised, unsupervised and self-supervised models, respectively. As shown in Fig.12, a consistent convergence for the curves of the angles can be observed and support our hypothesis that the dynamics learning procedure construct the common subspace gradually.

## A.4 THE NON-MONOTONIC TREND IN THE FIRST EPOCH OF COMPARISON OF ANGLES BETWEEN MODEL AND RAW DATA $\mathcal{P}$-VECTORS

We also explore the construction procedure for the common subspace share between feature vectors and the raw data during the training process. Experiments are carried out to compare the space of models and raw data $\mathcal{P}$-vectors on the training dataset. As shown in Fig.13, we observe a non-monotonic trend that the angle first rises with the random initialization and drop down. The angles keeps the approximately monotonically decreasing and converging to small values in following training epochs. The experiments shows consistent result on both CIFAR-10 and 100 dataset. Note that we follow the default random data augmentation policy to pre-process the training dataset.

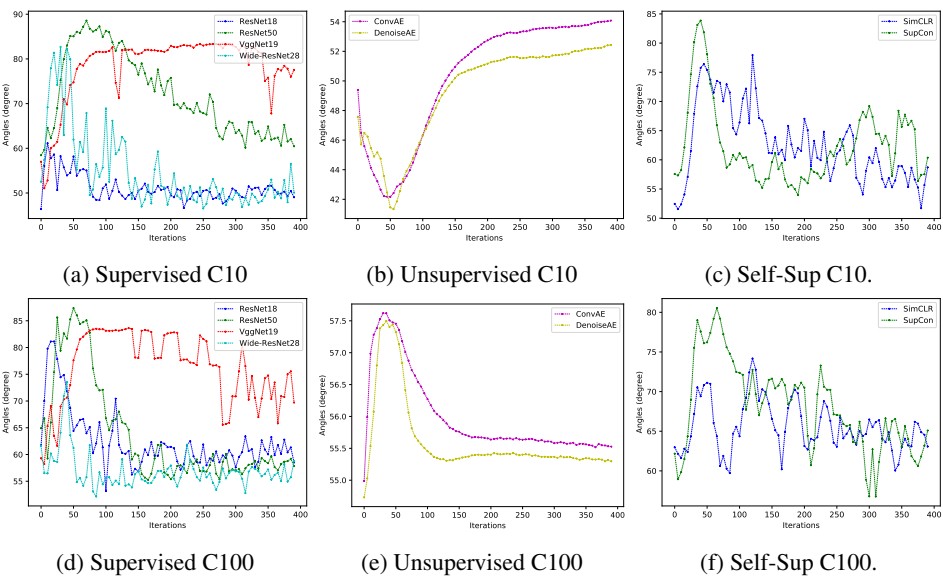

Figure 13: Angles between the $\mathcal{P}$-vectors of the training and well-trained models over the number of iterations in the first epoch using CIFAR-10 /CIFAR-100.

## A.5 CASE STUDIES ON THE ANGLE VARIETY BETWEEN MODEL AND RAW DATA FOR EACH LAYER

To explore the dynamic variation process by zooming in to every layer variation in each epoch, we perform the case study using Resnet-18 structure on CIFAR-10 dataset. As shown in Fig.14, we give the angles according to layers of 5 different epochs, where the x-axis indicating the indices of residual blocks in the network structure and y-axis refers to the angles between the model checkpoint and raw data $\mathcal{P}$-vectors. We observed that in early training stage, the angles between the model checkpoint and raw data $\mathcal{P}$-vectors keeps an increasing manner when the features passing though layers and turn into a decrease trend towards the stacked layers in the late training stage. This set of experiments further support our hypothesis that the dynamics learning procedure construct the common subspace gradually through training process.

## A.6 DISTRIBUTION OF VALUES IN THE $\mathcal{P}$-VECTOR

Please refer to Figure 15 for the results of experiments carried out on CIFAR-10 and CIFAR-100 datasets using ResNet-50.

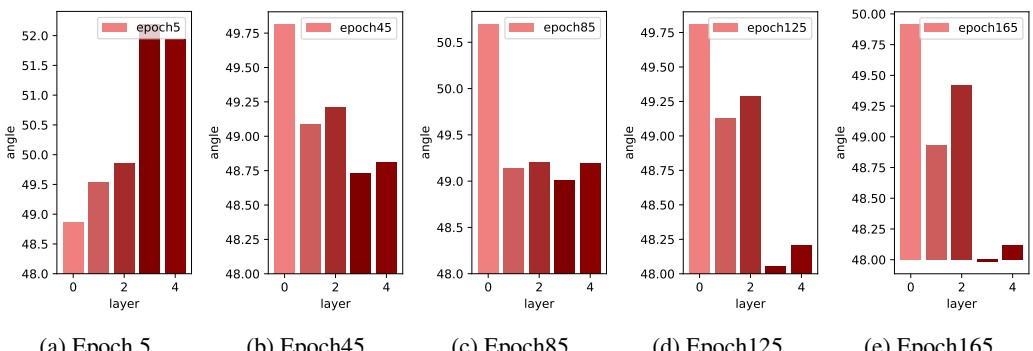

(a) Epoch 5.     (b) Epoch45.     (c) Epoch85.     (d) Epoch125.     (e) Epoch165.

Figure 14: Angles changes through layer between the $\mathcal{P}$-vectors of the training and raw data over the number of iterations in the first epoch using CIFAR-10.

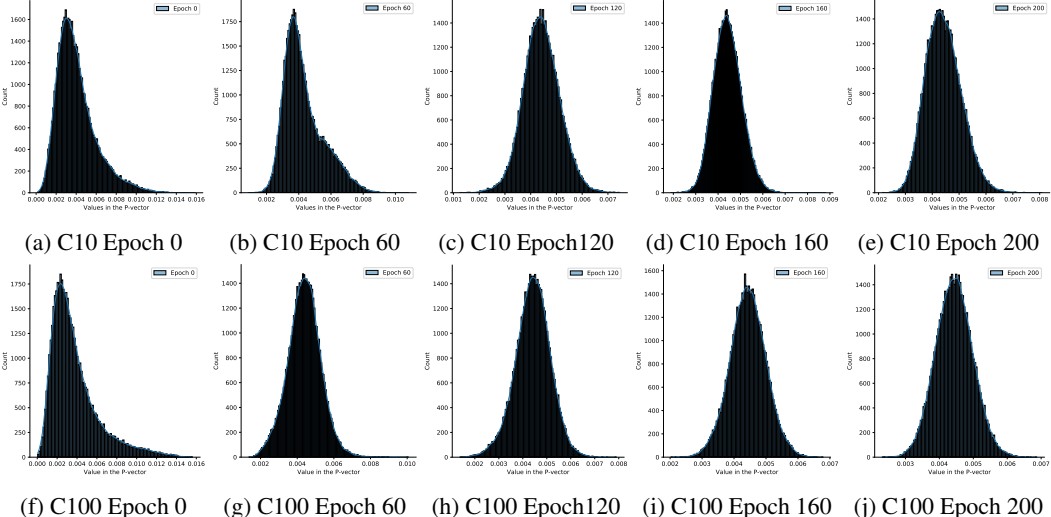

(a) C10 Epoch 0   (b) C10 Epoch 60   (c) C10 Epoch120   (d) C10 Epoch 160   (e) C10 Epoch 200

(f) C100 Epoch 0   (g) C100 Epoch 60   (h) C100 Epoch120   (i) C100 Epoch 160   (j) C100 Epoch 200

Figure 15: Frequency distributions of the P-vector values over training epochs.

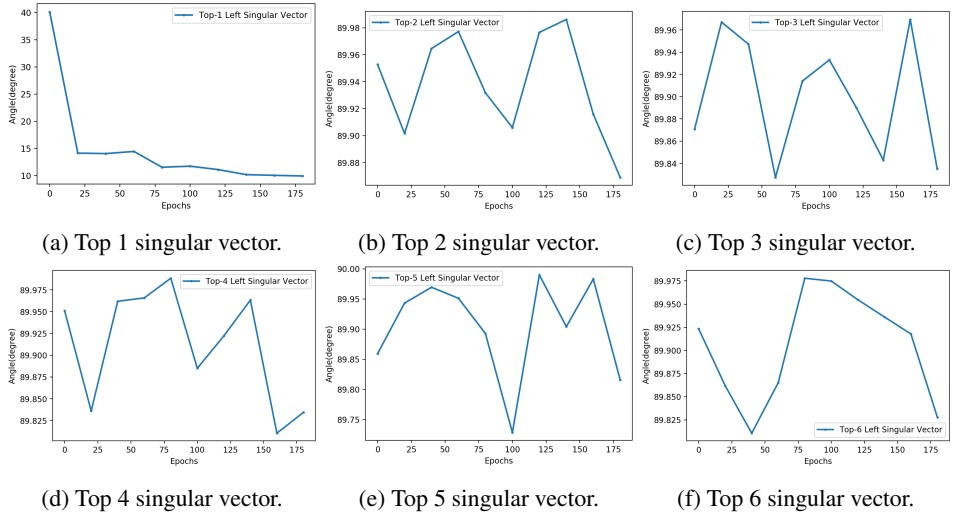

Figure 16: Angles between the top-k left singular vector ($k = 1$ is the $\mathcal{P}$-vector) of the training and well-trained models over the number of epochs in the training process (Resnet-50, CIFAR-10). Note the first plot point refers to the feature matrix after trained for one epoch.

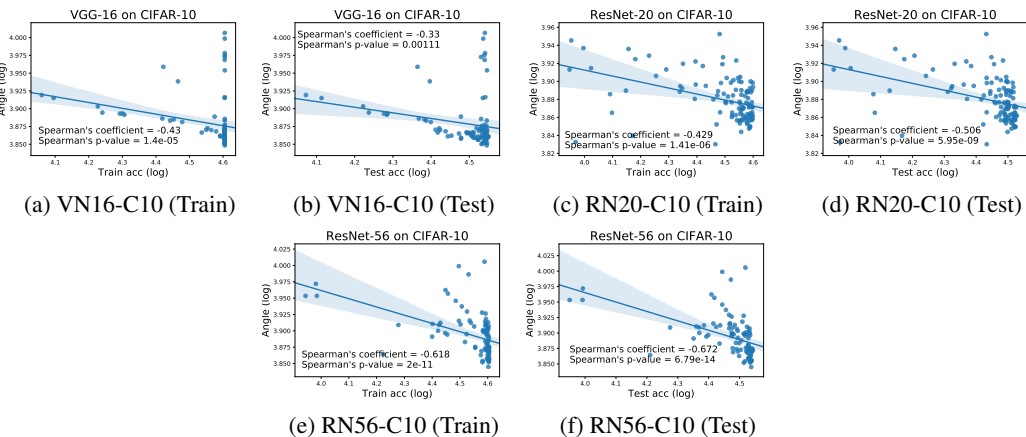

Figure 17: Log-Log Plots: Strong and consistent correlations between the model performance (training and testing accu racy in log range) and the angels (in log range) between model and data $\mathcal{P}$-vectors using CIFAR-10 datasets. VN: VggNet, RN: ResNet, C10: CIFAR-10.

## A.7 NO CONVERGENCE FOUND IN COMPARISONS BETWEEN THE TOP SINGULAR VECTORS OTHER THAN THE $\mathcal{P}$-VECTORS

Please refer to Figure 16 for the results of experiments carried out on CIFAR-10 datasets using ResNet-50.

## A.8 LOG-LOG PLOTS THAT CORRELATE THE $\mathcal{P}$-VECTOR ANGLES AND THE MODEL PERFORMANCE

Please refer to Figure 17 for the log-log plot of the results based on on CIFAR-10 datasets.

