# OpenReview forum: "Empirical Studies on the Convergence of Feature Spaces in Deep Learning"
_ICLR.cc/2021/Conference — Reject_

### Official Review · AnonReviewer4 · 2020-10-26
**Interesting hypothesis but results are not convincing enough**

**Rating:** 3
**Confidence:** 4

**Review:**

This paper studies the top singular vector of the feature space learned by supervised and unsupervised deep learning models on CIFAR datasets. The hypothesis of converging feature spaces is interesting (converging both in terms of different models, and in terms of training epochs), but the conclusion from the current experiment results is overstretching.

1. While the authors emphasize the convergence of subspaces, the P-vector defined in the paper is actually the top singular vector of the feature space, so it's actually about the convergence of the 1-dimensional principal subspace. A subspace refers to an arbitrary dimensional space in general. In the context of SVD, the literature often studies the top-$k$ dimensional subspace, which is represented by the $k$ top singular vectors, and the approximation error of the top-$k$ dimensional subspace: $E=\|X - U_k \Sigma_k V_k^T\|_F^2$, where $X$ would be the feature matrix in this paper, and $U_k, V_k$ are the first $k$ columns in the result of SVD. The authors didn't measure $E$, so the readers won't know how well the top-1 dimensional subspace represents the feature matrix. I recommend looking at $E$ as a function of $k$, and use some criteria to determine how closely you want the subspace to approximate the feature matrix. For example, we can say we want to keep the top-$k$ dimensional subspace such that $E < 0.1 \|X\|_F^2$. This way, you can rule out the possibility that the P-vector is a trivial vector that every model will converge to.
(As an analogy for a trivial vector, we can consider the top-1 eigenvector of the similarity matrix defined in the classical spectral clustering method called Normalized Cut. No matter how the edge weights in a graph is defined, the similarity matrix used in Normalized Cut always has an all-one vector as the top-1 eigenvector.)
And to measure the angle between general subspaces, many methods are available including classical ones (e.g. Åke Björck and Gene H. Golub, Numerical Methods for Computing Angles Between Linear Subspaces, 1973).

2. This paper tries to emphasize the P-vectors found in the features from different deep learning models are very close (for example, "no matter what type of DNN architectures or whether the labels have been used to train the models, the P-vectors of different models would converge to the same one"). Actually it seems the angle typically converges to 10 to 20 degrees. It may be better to lower the tone, or quantify better (compared to the angles obtained by ..., the angles between P-vectors are smaller).

3. The data in Fig. 7 looks quite noisy, though p-value shows statistical significance of the correlation. p-value can guide our findings but is not always meaningful. For example, comparing Fig.7(e) and Fig.7(l), we may argue the latter has a better correlation but the former has a much smaller p-value. It seems the very small p-value in Fig.7(e) results from some outliers. Intuitively I don't quite understand why the raw data and the features should have a correlated linear principal subspace, given that the neural network layers that generate the feature from the data are highly nonlinear.

The only convincing data I found is in Table 1, which shows P-vectors can serve as an indicator of the model performance. But overall the readers would need more evidence as explained in #1 above.

---

> ### Author Response · Authors · 2020-11-18
> **Response to ICLR 2021 Conference Paper1111 AnonReviewer4 Continued**
>
> Even you conclude the P-vector is trivial upon the additional results above and figures in the appendix, we still make contribution in this work–it might be the first evidence on the triviality of principal subspace of features learned by DNN or well-trained DNNs would converge to have a trivial principal subspace in their feature learning. Many thanks for mention the existing work in measuring the angle between principal subspaces. We will include these work in the discussion.
>
> We really appreciate your guidance here. We will significantly extend the methodology section to first analyze the singular value distribution of feature matrices, explained variances, and the reconstruction error of approximation using the top-k singular vectors, prior to introducing our observations! Many thanks!
>
> Explained Variances of top-k singular vectors:
>
> |  k   | Ratio |
> |  ----  | ----  |
> | 1  | 0.5674805 |
> | 2  | 0.6489089 |
> |3 | 0.7067079|
> |4 | 0.75878125|
> |5 | 0.8063708|
> |6 | 0.8489084|
> |7 | 0.8891225|
> |8 | 0.92527133|
> |9 | 0.9577149|
> |10 | 0.98870885|
> |11 | 0.9896392|
> |12 | 0.9902976|
>
> Many thanks for your advices. We will lower our tone in the formal revised manuscript. Indeed, cosin(10$^\circ$)=98.5\% and cosin(20$^\circ$)= 94.0\%, are quite small when we treat the cosine measure in a correlation sense. All in all, in our formal revised manuscript, we will conclude the “converging trends” of the angle between P-vectors, and emphasized that it would not converge to zero, but to a much smaller angle.
>
> Many thanks for your question on the correlation between features and raw data in their linear principal subspace. First of all, this is just our empirical observation and we report it. Latter, to avoid possible affects due to the outliers, we tested the hypothesis using Spearman’s correlation to correlate the angles and training/testing accuracy, which considers the order of the data point in the samples rather than the value of them. In this way, few outliers would not dominate the correlation analysis. In the current modified manuscript, we take the log-log plots, where we can see the same phenomena. Thus, we don’t believe a few outliers would affect our conclusion here. Though the overall DNN is nonlinear, many previous work [1],[2] also demonstrates the local linearity or piecewise linearity of DNN with certain activations. We believe it is the linear subcomponents of the DNN (e.g., a significant first-order term in the Taylor expansion) that “leaks” certain information about the raw data through a weak linear transform. In this way, the correlation between features and raw data in the linear principal subspace characterizes “how linear is the feature extractor of a DNN model” which should relates to the (generalization) performance.  Of-course, it is all of our intuition. We didn’t hope to claim it in the manuscript. To address your comment, we will discuss this issue in the formal revised manuscript to elaborate our intuition well.
>
> We really appreciate your comments. We hope to could get a chance to be shepherded and improve the manuscript. Please feel free to comment on the discussion thread, especially when we made anything wrong. Many thanks!
>
> [1] Zhang X, Wu D. Empirical Studies on the Properties of Linear Regions in Deep Neural Networks[C]//International Conference on Learning Representations. 2019.
>
> [2] Arora R, Basu A, Mianjy P, et al. Understanding Deep Neural Networks with Rectified Linear Units[C]//International Conference on Learning Representations. 2018.

---

> > ### Comment · AnonReviewer4 · 2020-11-19
> > **Thanks for the additional analysis**
> >
> > Thanks for the additional analysis on the reconstruction error. From Fig. 16 in the appendix (ignoring the data point at epoch 0), it seems that as the epoch increases, the weight of the matrix is spread to more dimensions, because the reconstruction error using the top-1 singular vector slightly increases. I agree it's not easy to say the top-1 singular vector is trivial or not. As mentioned in my original comment, you have the choice to define which level of reconstruction error works for you (i.e. how much variance you want to capture in the principal subspace, which in general has more than 1 dimensions). For example, maybe you believe $E < 0.6 * \|X\|_F^2$ is good enough, then the top 1-dimensional principal subspace is good to use; if you prefer to have $E < 0.2 * \|X\|_F^2$, adding more singular vectors to the subspace will be necessary. Essentially it will be good to quantify how much variance the principal subspace captures, and the principal subspaces capturing the same percent of variance of different feature matrices will be made comparable. Then the scope of this work can be accurate. Looking forward to the discussion of measuring the angle between principal subspaces of general dimensions.
> >
> > By the way, for Fig. 15 it may give more insight by normalizing the singular values: $\sigma_i / \sqrt{\sum_j{\sigma_j^2}} = \sigma_i / \|X\|_F$, because $\|X\|_F$ of feature matrix $X$ increases during training, empirically.
> >
> > Thanks for the explanation of local linearity. It makes sense to me now.

---

> > > ### Author Response · Authors · 2020-11-20
> > > **Additional Response to ICLR 2021 Conference Paper1111 AnonReviewer4**
> > >
> > > Many thanks for your further comments. Sorry for the late reply.
> > >
> > > **Only P-vector converges in our empirical study.** You mentioned that when the reconstruction error is not good enough according to different criterion, adding more singular vectors to the subspace will be necessary. Actually, we have conducted experiments using top-k left singular vectors (k=1 is P-vector). However, in contrast to P-vector, the angles between other top singular vectors do not show a trend of convergence in the training process. The angles between the top-k left singular vectors (>1) of the training and well-trained models are close to be orthogonal. Besides, the angles jitter intensively over the number of epochs in the training process, as shown in the revised appendix Fig.17.
> > >
> > > A possible explanation for the inconsistency of other top singular vectors is the “mismatching of indices” in the SVD of feature matrices. While the top-1 singular vectors always dominant the feature space, the 2nd, 3rd, … top singular vectors may have different permutations as the singular value distribution shifts and alters in the training process. Thus, adding more singular vectors to the measurement would be difficult (maybe using optimal transport is a solution to compare two sets of top singular vectors between models, however it is another topic of research). Therefore, only the P-vector is principal and definitely matched cross feature spaces and that is the reason why only P-vector is used in our empirical studies.
> > >
> > > Thanks again for the comments. To address your concerns, we will include a discussion on the angles using different dimensions of top left singular vectors, in the formal revised manuscript. Again, please feel free to ask any more questions or make any more comments. It’s our pleasure to have further discussions.

---

> ### Author Response · Authors · 2020-11-18
> **Response to ICLR 2021 Conference Paper1111 AnonReviewer4**
>
> Many thanks for the review and constructive comments. We must say your comments are quite helpful to shepherd us for approaching a better work. Actually, we have upload a modified version of the manuscript, with a new appendix including additional figures/tables/evidences to address some of your concerns. We are working hard to revise the manuscript accordingly to address all your concerns and fix all language and presentation issues. For the formal revised manuscript, we will upload it before the end of rebuttal period.
>
> On the triviality of the P-vector. Many thanks for your comments. We also have concerns on the validity of our observations. With your comments, we preformed SVD on the feature matrices (#samples $\times$ #features = 50,000 $\times$ 256) obtained using ResNet50 and CIFAR-10 data. We attached the result in appendix of current modified manuscript. The result shows the feature matrix is extremely spiked (with use log x-axis in the figure). Through comparing the singular value distributions obtained in different training epochs, we can observe the progression of the feature matrices during the training procedure. For both initial model and well-trained model, the most of energy is distributed in the first 10 singular vectors for the well-trained model. In the explained variance analysis, the variance ratio of top-10 singular vectors could achieve 98.8\%, while the variance ratio of top-1 singular vectors (i.e., P-vector and the corresponding right singular vector) could achieve 56.8\%.
>
> Further, we follow your insightful instruction to analyze the low-rank approximation to the feature matrix using the principal singular vectors. We first estimate the reconstruction error of feature matrix using the P-vector and the corresponding top right singular vector with the singular value. We have included a new figure on reconstruction errors using P-vectors in the appendix of current modified manuscript. The reconstruction error grows from 40\% to 60\%, i.e., $E_1=0.4*|X|_F^2$ in the begin of training procedure while $E_1=0.6*|X|_F^2$ for well-trained one, using the single P-vector. It coincides our observation in the singular value distributions. After all, we are not an expert on the determining the triviality of singular vectors. However, no matter whether the P-vector is trivial or not, we believe (1) the P-vector well represents the feature matrix in a data-driven fashion, (2) the P-vector contains information about the distribution of samples in the feature space (see the explained variance analysis), and (3) the singular value distribution of the feature matrix would vary over the training epochs while the distribution would become flatten among the top-10 singular vectors in our experiments. Please advise us if we make anything wrong here.

---

### Official Review · AnonReviewer3 · 2020-11-01
**Marginally below acceptance threshold**

**Rating:** 5
**Confidence:** 3

**Review:**

Summary:

This paper has a closer look at the distributions of samples in the feature space by utilizing P-vector to analyze principal subspace. According to their empirical studies, the authors concluded that the feature spaces learned by different deep models with the same dataset would share common principal subspaces for the same dataset. It will not be affected by DNN architectures or the usage of labels in feature learning. Only the training procedure gradually shapes the feature subspace to the shared common subspace.

-----------------------------------------------------------------------

Reasons for score:

The paper explores a new question and gives some interesting conclusions. But my major concern is its empirical studies cannot support the findings well. Besides, there are few discussions to provide the readers with some insights. I hope the authors carefully consider how to enhance this paper and make the conclusions more convincing.

-----------------------------------------------------------------------

Pros:

1. The paper explores a new question and gives some interesting conclusions.

2. The proposed metric is simple and easy to follow. The authors also attached the source code for reference.

3. The usage of the P-vector for predicting generalization achieves promising results.

-----------------------------------------------------------------------

Cons:

1. Why can the similarity of P-vectors be used to indicate the similarity of two distributions in the feature space?
2. I would like to know how many trials it takes to plot similarity figures (e.g., Figure 1 (a)-(b)). It would be better to try many times and give the mean and variance to avoid coincidence. Besides, will other parameters such as the number of samples and dimensions of features affect Hypothesis I?
3. The authors mentioned that the reference model used in Figure 4 (a)-(c) is Wide-ResNet28 trained with 200 epochs under suggest settings. But, the plot of Wide-ResNet28 in Figure 4 (a) is weird. It cannot converge to zero.
4. I would like to know why most models (such as Figure 4) cannot converge to zero after about 200 epochs training, and the angles are approximately 10 degrees. In other words, is there exists a threshold after which we can think the compared two models have a common subspace?
5. Why the P-vector can be used to predict the generalization?

-----------------------------------------------------------------------

Questions during the rebuttal period:

Please address and clarify the cons above.

-----------------------------------------------------------------------

Some typos:
(1) Figure 5, Figure 9, the xaxis's titile should be iterations rather than epochs.

---

> ### Author Response · Authors · 2020-11-18
> **Response to ICLR 2021 Conference Paper1111 AnonReviewer3 Continued**
>
> Many thanks for your comments on the definition of “convergence”. We agree with you that the angles between P-vectors here usually cannot converge to zero after 200 epochs training, and the angles remaining are around 10$^\circ$ degrees.  Actually, we use top left singular vector of the feature matrix (i.e., #samples $\times$ #features) as the P-vector. Thus, the number of dimensions of a P-vector is equivalent to the number of samples in a dataset. When we use CIFAR-10 or CIFAR-100 for experiments, the P-vectors should be with 50,000 dimensions. Please note that the high-dimensional random vectors are tending to orthogonal when they are not correlated. In our opinion, 10$^\circ$ degree is indeed quite small in such case, compared to the 80$^\circ$ to 90$^\circ$ in the begin of training procedure. Furthermore, cosine(10$^\circ$) =0.985, which is quite significant in similarity comparison. Thus, we believe the remaining angles would not hurt our claims. To address your comments, we will include a discussion on the definition of “convergence” in the formal revised manuscript.
>
> Many thanks for your comments on the relation between P-vector and “generalization”. Indeed, we estimate the top left singular vector of the raw data matrix as the data P-vector and use the angle between the data P-vector and the P-vector (obtained from the feature matrix of a model) to predict performance of the model. A small angle between the data P-vector and the P-vector well demonstrates the correlation or divergence between the distribution of samples in the raw dataset and the distribution of samples in the feature space, through comparing their principal subspaces. Our intuition is that when the principal subspace of the raw dataset is close to the feature one, the CNN model would preserve more information about the data distribution (even though the models are intensively parameterized and trained), demonstrate higher linearity (as the principal subspaces of features should equivalent to the principal subspace of raw data when only linear transform applied to the data), are supposed to be with better generalization performance. We are inspired by some work on the local linearity of CNN, such as [1] and [2]. After all, we only hope to report our observations as a potential application of P-vector, while we don’t intend to over-claim the effectiveness of P-vector for model selection and/or generalization prediction. To address your comments, we will discuss these issues in the formal revised manuscript.
>
> Please check the appendix of current modified manuscript for additional examples, observations, and evidences. Again, many thanks for your review and encouraging comments. We will address all your concerns and fix language issues in the revised version. Please feel free to comment on the thread of discussion timely and shepherd us for improving the manuscript.
>
> [1] Zhang X, Wu D. Empirical Studies on the Properties of Linear Regions in Deep Neural Networks[C]//International Conference on Learning Representations. 2019.
>
> [2] Arora R, Basu A, Mianjy P, et al. Understanding Deep Neural Networks with Rectified Linear Units[C]//International Conference on Learning Representations. 2018.

---

> ### Author Response · Authors · 2020-11-18
> **Response to ICLR 2021 Conference Paper1111 AnonReviewer3**
>
> Many thanks for the review and encouraging comments. Actually, we have upload a modified version of the manuscript, with a new appendix including additional figures/tables/evidences to address some of your concerns. We are working hard to revise the manuscript accordingly to address all your concerns and fix all language and presentation issues. For the formal revised manuscript, we will upload it before the end of rebuttal period.
>
> The P-vector, as the top left singular vector of the feature matrix (#samples $\times$ #features), indicates the principal subspace of the samples encoded by CNN features (namely feature vectors of the samples). According to the definition of SVD and PCA, in our settings, the principal component (top-1) projects the feature vector of every sample into a 1-dimesnional principal subspace that the variance of these data points is maximized. The value of every sample in the P-vector here refers to the position of the sample in such 1-dimensional space. In this way, we believe the principal subspace characterizes the distribution of the samples in the feature space, and the P-vector represents such distribution (at least partially). Thus, the comparison between the P-vectors is meaningful.  In addition, though P-vector only takes the top-1 left singular vector of the feature matrix, comparisons between the P-vectors avoid a lot of technical problems, such as dimensionality mismatching issues for CNN features from different architectures. To address your concerns, we will discuss this part with running examples in the formal revised manuscript.
>
> In Figure 1(a)—(b), to alleviate the influence of random initialization, we carry out and repeat the experiments using 5 random seeds independently. In every experiment, we train the DNN models of various architectures, and compare the P-vectors between any two architectures using cosine measure. Finally, we average the cosine measures obtained in the 5 independent trials. Many thanks for your suggestion, we will include the statistics on mean and variance. Furthermore, our experiments half dozens of architectures, three datasets of various sizes, three learning tasks of various losses, and 5 random seeds, all based on default hyper-parameters/training settings, we have not found “parameters such as the number of samples and dimensions of features affect Hypothesis I”. To address your concerns, we will include a discussion on the internal and external threats to validity in the formal revised manuscript.
>
> Many thanks for your note on the jitters in the curve of Wide-ResNet28. In Figure 4 (a) -(c), we compare the P-vectors of various models obtained during the training procedure to the P-vector of the well-trained Wide-ResNet28 (200 epoch). We can see the curve of Wide-ResNet28 (P-vectors between the training Wide-ResNet28 versus well-trained Wide-ResNet28) with small jitters around 10$^\circ$, while rest lines look smooth. Actually, similar observations could be obtained in the P-vector comparisons between the training Wide-ResNet28 and well-trained ConvAE (Figure 4 (d)) and the P-vector comparisons between the training Wide-ResNet28 and well-trained SimCLR (Figure 4 (g)). We indeed carried out the experiments using 5 random seeds independently. In every experiment, we obtain the curve of cosine measures between P-vectors of any two models and average the cosine measures to plot Figure 4. We can see the P-vectors of Wide-ResNet28 over the training epochs are with higher perturbations than others. We conclude it is a characteristic of Wide-ResNet28, as such architectures is wide, over-parameterized, and easy to be over-fitted. Actually, the CNN feature extractor of Wide-ResNet28 outputs feature vectors with 1280 dimensions, while other networks usually generate features with extremely less dimensions (e.g., 256 for ResNet). In this way, the feature vectors obtained by Wide-ResNet28 are easier to incorporate noise or redundant information during the training procedure. They are much more stochastic than other modes. Thus, even we compare the P-vector between the training and well-trained models of Wide-ResNet28, the difference might be higher than the rest models. To address your comments, we will discuss these issues in the formal revised manuscript.

---

### Official Review · AnonReviewer1 · 2020-11-02
**Interesting experiments into neural networks behaving similarly**

**Rating:** 6
**Confidence:** 4

**Review:**

The authors identify an interesting empirical phenomenon: across a range of network architectures and training approaches (supervised, unsupervised, auto-encoders), the feature spaces identified by these networks are similar. The authors introduce a specific way to summarize the feature space of a network as a vector (the top-left singular vector of the num_examples x num_features matrix) and show that these vectors are highly correlated across networks. In addition, the authors show that the features spaces become more similar throughout training and are predictive of the generalization performance of a neural network.

The paper presents purely experimental findings, but the experiments are sufficiently broad (e.g., covering different training approaches and datasets) so that this is not a shortcoming. Investigating potential theoretical models or more models and datasets could be fruitful directions for future work.

Aspects of the presentation in the paper could be improved (see the comments below). Overall I still recommend accepting the paper.


Additional comments:

- Many plots have labels that are too small to read. I strongly encourage the authors to produce more readable plots.

- Have the authors explored visualizations of the P-vectors? For instance, what is the ordering of training examples induced by the P-vectors?

- What is the "SupCon" method? Do the authors have a hypothesis for why it behaves different from the other methods w.r.t. P-vector angles?

- Have the authors experimented with training approaches that explicitly encourage small angles between model and data P-vectors?

- The paper would benefit from a thorough editing pass to fix typos and improve clarity. The structure of the paper is well-organized, just some sentences are hard to parse.

- When abbreviations like "AE" or "CNN" are used for the first time, it is generally good to write them out.

- Why does the paper sometimes use angle and sometimes use cosine of the angle? It could be better to use one of the two consistently.

- Should the x-axis labels in Figure 5 be "training steps" instead of "epochs"?

- There is too little vertical space separating the caption of Figure 7 from the text below.

- Typos:
  * Introduction: "To better euclid"
  * Section 3: "various architectures amd different training paradigms"
  * Section 5: "an data, We carry out"  (capitalization)
  * Section 5: "expect" -> "except"

---

> ### Author Response · Authors · 2020-11-18
> **Response to ICLR 2021 Conference Paper1111 AnonReviewer1**
>
> Many thanks for the review and encouraging comments. Actually, we have upload a modified version of manuscript, with a new appendix including additional figures/tables/evidences to address some of your concerns. We are working hard to revise the manuscript accordingly to address all your concerns and fix all language and presentation issues. For the formal revised manuscript, we will upload it before the end of rebuttal period.
>
> Actually, in P-vector computation, we use the index of every sample in the training/test datasets as the indices of dimensions in the P-vector. In appendix of the current modified manuscript, we have provided the visualization of the P-vector, including a figure on the P-vector values versus the indices of samples, figures on the frequency of the P-vector values (counts versus the P-vector values), and a figure on the smoothed density of the P-vector values (probability density versus the -vector values). The feature matrices are obtained through training a ResNet50 model using CIFAR-10 datasets, the feature matrices of 0th epoch (obtained by random initial weights), 60th epoch, 120th epoch, and 200th epoch (well-trained) have been used for plots. In the formal revised manuscript, we will include all results based on the three datasets.
>
> SupCon refers to the supervised contrastive learning [1]. We include SupCon together with SimCLR [2] as two typical algorithms for self-supervised training. Generally, self-supervised learning, especially SimCLR [2], aims at improving the training procedure of deep neural networks with self-supervised contrastive loss.  While SimCLR [2] trains CNN feature extractors without the use of label information, SupCon [1] extended the self-supervised paradigm using supervised contrastive learning (which is based on the labels). To make it clear, in the revised manuscript, we will address this issue when we introduce the self-supervised learning paradigms.
>
> We didn’t incorporate any algorithms, regularizers, or any treatments to make the angles between P-vectors smaller. All algorithms used for DNN training here were based on the open source implementations that are available online. The goal of our research is to investigate the similarity between P-vectors for DNN trained using the same dataset with various architectures/tasks. All experiments were carried out to follow the standard implementation and operations.
>
> Again, many thanks for your review and encouraging comments. We will address all your concerns and fix language issues in the revised version. Please feel free to comment on the thread of discussion and timely shepherd us for improving the manuscript.
>
> [1] Khosla P, Teterwak P, Wang C, et al. Supervised contrastive learning[J]. arXiv preprint arXiv:2004.11362, 2020.
>
> [2] Chen T, Kornblith S, Norouzi M, et al. A simple framework for contrastive learning of visual representations[J]. arXiv preprint arXiv:2002.05709, 2020. MLA

---

### Author Response · Authors · 2020-11-25
**Summary of Changes (Part 2)**

6.	**Correlation and Significance Test.** In fact, to avoid the dominance of some outliers in the correlation study, we performed the significance test using Spearman’s correlation coefficients, where we majorly correlate the rank of model performance and the rank of P-vector angle among a large number of trained models. To make it even clearer, we also analyze the correlation in logarithmic scale i.e., “accuracy (log) versus P-vector angles (log)”. As the Spearman’s correlations (i.e., rank information) are used, the results of significance tests are the same. We truly understand that it is inappropriate the compare the exact values of two correlation coefficients or P-values here, as the numbers of models in comparisons here are still small. Thus, we revise Section 5 of Page 8 in the revised manuscript to address the concerns and include the log-log plots in Appendix (A.8 of Page 16 in the revised manuscript).

7.	**Language and figure issues.** Many thanks for pointing out the language issues, such as the use acronyms (such as “AE” or “CNN”) before defining them and inappropriate use of x-axis titles (e.g., epochiteration). We have fixed them and will continue improving the writing of this manuscript.

Many thanks for Reviewers’ time and patience in reviewing this paper and shepherding us to improve the manuscript. It is quite an enjoyable period for us to discuss and receive instructions from reviewers in these days. We have uploaded a revision to the manuscript. Hopefully, this manuscript could receive your full consideration for acceptance. Any help from you is highly appreciated. Thank you!

---

### Author Response · Authors · 2020-11-25
**Summary of Changes (Part 1)**

We thank AC’s efforts in organizing the reviews. Many thanks for reviewers. We do appreciate your efforts in helping us improve the manuscript. Reviewers raised major concerns from the following perspectives.

1.	**Visualization of P-vector.** In the revised manuscript, we include two examples to visualize P-vectors using CIFAR-10 and CIFAR-100 datasets. Specifically, P-vector is an extremely high-dimensional unit vector, and every sample refers to a dimension in the vector. We are interested in the value on every dimension in the vector. We present the distribution of values in the P-vector in Figure 4 of Section 4 in Page 5, where we plot the smoothed probability density of the distributions obtained in various training epochs. In Appendix (A.6 in Page 14), we present the raw frequency of values in the P-vector. From these visualizations, we can clearly understand the shape that values in a P-vector distribute and how they change throughout the training procedure.

2.	**Experiment Settings and SupCon.** Actually, for the experiment results reported in Figures 1, 5, 6, 7 and 8, we carried out experiments using 5 independent trials with different random seeds and averaged the results. The SupCon’s behavior is significantly different from another contrastive learning approach SimCLR, as it incorporates both self-supervised (contrastive) and supervised learning loss. To address your comments, we have included the introduction to the experiment settings in the caption of Figure 1.

3.	**Definition of Convergence.** In our experiments, no model’s P-vector angles could strictly converge to zero. In this way, we lower tone to claim the converging trend of angles to smaller ones. Further, we define the phenomena of “convergence” in the footnote of Page 3 in the revised manuscript, where we clearly state that “In our research, we name convergence as the decreasing trend of P-vector angles from a larger one to a smaller one (e.g., $10^\circ$ for supervised CNN classifiers and SimCLR, $<10^\circ$ for ConvAEs/DenoiseAEs, and $30^\circ$ for SupCon) over training epochs. For reference, $Cosine(10^\circ)=0.985$ is close to 1.0.

4.	**Why P-vector could be used to predict performance.** All reviewers seem interested in the motivation that we correlate the angles between model and data P-vector and the performance of models. Especially, the angles are estimated using the training dataset without validation information, but they correlate to the testing accuracy. We are motivated by the earlier observations that well-trained DNN models are locally or piecewise linear (Zhang & Wu, 2020; Arora et al, 2018). Considering the DNN model as a mixture of linear and nonlinear transform, when the linear part is significant in the output of feature extractor, the subspace of feature matrix should be close to the subspace of data matrix. In the original submission, we introduce our intuitions in paragraph of “Hypothesis” in Page 1. To further address your concerns, we elaborate our intuition in Section 5 of Page 8 in the revised manuscript and discuss this issue again in the Conclusion.

5.	**Triviality/significance of P-vectors.** We appreciate reviewers’ comments to help us rethink more about the capacity of P-vectors. Under the guidance, we carried out the Singular Value Decomposition (SVD) Analysis on the feature matrices obtained from ResNet-50 using both CIFAR-10 and CIFAR-100 datasets. The results are presented in Figure 3 of Page 5 in the revised manuscript, where we present the distributions of singular values and how the distribution changes throughout the training procedure. Cliff patterns have been observed in the singular value distributions. Only few (less than 10) singular values are significant, while the rest are all close to zero. We also carried out explained variance analysis. It shows that P-vector and the corresponding top right singular take 50% of total variances, while the second top singular vectors take less than 10%. These evidences suggest that the P-vector could represent the subspace of features learned. Furthermore, in Appendix (A.7), we also include the analysis results using top-2, 3, 4, 5, and 6 left singular vectors, no converging trends could be observed when comparing these vectors in every training epoch and well-trained ones. Many thanks for the comments.

---

### Decision · Program_Chairs · 2021-01-07
**Final Decision**

**Decision:**

Reject

**Comment:**

This paper presents an intriguing empirical phenomenon in deep learning. They train a variety of architectures for different tasks using different datasets and study the relationship between the learned representations. In particular they collect the representations into a large matrix and take the top left singular vector and measure the cosine of the angle. They show that it is much smaller than one might expect, about 10 degrees or so, and has an approximate monotonicity property as the network is being trained although it does not seem to converge to zero. Moreover this measure also correlates with performance.

The reviewers had divided opinions on this paper. On the one hand, the range of experiments is impressive and truly demonstrates that this is a pervasive phenomenon. On the other hand, it is not so clear what it means. In particular, suppose we have a collection of graphs which have close to the same degree distributions. If we take the top left singular vectors of all the adjacency matrices, they would also have low angles between them. While this is a very different setting and there is no analogy between the experiments in this paper and this toy model, it does raise philosophical questions about whether the phenomenon is meaningful or is a byproduct of something else about the data. This may be a challenging question to answer, but one reviewer brought up a natural next step: One could measure the principal angle between the subspace of the top k left singular vectors across experiments for larger values of k. The authors do bring up the point that the spectrum decays very quickly, so it could be that beyond a certain point the singular vectors behave somewhat randomly.